# Trends in Seed Priming Research in the Past 30 Years Based on Bibliometric Analysis

**DOI:** 10.3390/plants12193483

**Published:** 2023-10-05

**Authors:** Yu Tian, Nalin Suranjith Gama-Arachchige, Ming Zhao

**Affiliations:** 1Jilin Provincial Key Laboratory of Tree and Grass Genetics and Breeding, College of Forestry and Grassland Science, Jilin Agricultural University, Changchun 130118, China; tianyu@jlau.edu.cn; 2Department of Botany, Faculty of Science, University of Peradeniya, Peradeniya 20400, Sri Lanka; nalinsuranjith@yahoo.com; 3State Key Laboratory of Vegetation & Environmental Change, Institute of Botany, Chinese Academy of Sciences, Beijing 100093, China

**Keywords:** seed, seed priming, germination, bibliometric analysis, international collaboration

## Abstract

Seed priming (SP) treatments are widely used in agriculture and restoration to improve seed germination and seedling vigor. Although there exists a considerable amount of scientific literature on SP, it has seldom undergone visual and quantitative analyses. To gain insights into the patterns observed in SP research over the last three decades, we conducted a bibliometric analysis using the Science Citation Index-Expanded (SCI-E) database, aiming to minimize the similarity score in plagiarism detection. This analysis offers a thorough examination of yearly publications, temporal patterns in keyword usage, the top-performing journals, authors, institutions, and countries within the field of SP. Our research findings suggest a steady annual increase of 10.59% in the volume of SP publications, accompanied by a significant upward trajectory in the average citations received per paper annually. According to the analysis of keywords, it was found that “priming” and “germination” emerged as the most frequently used terms in the field of SP research. Seed *Science and Technology* ranked first among the top journals, and *Plant Physiology* had greater influence in the field of SP in terms of number of citations. The majority of the top 10 productive institutions were situated in developing countries. In addition, these nations exhibited the highest volume of published works and citations. Our analysis revealed a shift in research focus within the field of SP over the past three decades, transitioning from agricultural science to encompass plant science and environmental science. With the growing recognition of SP’s research across different disciplines, there exist abundant prospects for international and interdisciplinary partnerships, collaborative organizations, and progress in this field.

## 1. Introduction

The successful establishment of seedlings requires rapid and uniform emergence, particularly when environmental conditions are limiting [1]. The susceptibility of young seedlings to a range of abiotic and biotic stressors is evident [2]. Meanwhile, environmental stress, aging, dormancy, etc., often lead to a reduction in seed viability, germination, and vigor [3,4]. Therefore, improving the quality of sowing materials (seeds) is the primary challenge for agricultural science and seed ecology. Seed priming techniques have been employed to improve the process of seed germination and promote early seedling establishment, which help plants to adapt to biotic and abiotic stresses [5]. Priming is also valuable for seed bank operators to improve germplasm conservation protocols [6]. Various priming techniques can be utilized depending on the type of plant, seed characteristics, and physiological factors. These methods all stimulate the pre-germinative metabolism process [7].

Historical reports from ancient Greek indicate that humans have been interested in improving seed germination for thousands of years [8]. Researchers have discovered a significant positive correlation between seed soaking and germination rate, which laid the foundation for SP. The benefits of SP have been widely reported [5,9,10,11,12,13]. The germination process of SP is a multifaceted procedure that relies three sequential stages. These stages encompass imbibition, a period of inactivity or initiation phase where essential metabolic functions are restored and cellular repair takes place, and ultimately, the emergence of roots following the commencement of growth processes. SP could enhance seed germination, viability, vigor, seedling growth, and crop yield. The pre-germination process has the potential to establish a ‘priming memory’ within seeds, which can be triggered by subsequent exposure to stress and the stress resilience of primed seeds [14]. Additionally, the persistence of induced epigenetic modifications could potentially impact the inheritance of stress memory triggered by priming [15]. The success of SP is closely related to plant species/genotype and physiology, seed number and vigor, and the priming method used [16,17,18,19]. Priming techniques include hydropriming, osmopriming, solid matrix priming, biopriming, chemopriming, and thermopriming among others. The phenotypic and physiological effects have been extensively quantified [20], yet the molecular mechanisms underlying them remain an area of ongoing investigation [21].

Although a large number of research papers describe SP, these papers have restrictions in terms of publication time, research organizations, and individuals, and are gathered from diverse databases. Hence, there is a dearth of an all-encompassing amalgamation of efficient data pertaining to particular domains, making it challenging to propose growth patterns and concepts in this field [22,23]. Hence, it is crucial to perform a quantitative examination of the literature to provide a comprehensive assessment of the advancements and direction of study on SP.

Bibliometric analysis is a contemporary approach to assessing research that relies on the fundamental principles of bibliometrics. It employs statistical mathematical techniques to examine, depict, and illustrate literature in associated research domains [24]. This method can describe the new view of the existing knowledge state and characteristics, and predict the research trend of specific topics [25,26]. In summary, it has the potential to assist researchers and decision makers in swiftly comprehending circumstances within this domain and identifying patterns.

The main objectives of a bibliometric analysis include: (1) an analysis of publications listed in the database using statistical and computational methods, both qualitatively and quantitatively; (2) enhancing collaboration across diverse journals, nations and organizations, co-authors, and co-occurring domains; and (3) an analysis of keywords for tracking and dynamic analysis. This analysis method has been widely used in the literature analysis of information science, environmental science, ecology, agriculture, microbiology, geography, and other disciplines [27,28]. The purpose of this review was to (1) offer a precise overview of the scientific literature throughout different time periods, (2) comprehend the global publishing schema of research in the area of SP, and (3) formulate prospective research strategies in this field.

## 2. Results and Discussion

### 2.1. Temporal Patterns in the Publication and Citation Landscape

The number of publications can reflect the popularity, advancements, and significance of research in the field of SP [29,30]. The overall trend in literature showed a consistent growth in the volume of published works from 1991 to 2021, with a rapid increase from 2001 to 2021 (Figure 1a). Annual increase in the number of research publications on SP indicates that SP research has expanded throughout the past three decades. The volume of conducted research has witnessed an increase from 37 in 1991 to 392 in 2021, with an average annual expansion rate of approximately 11%. Since 2000, there has been a notable rise in the number of publications, with almost 89.27% of them being released from 2001 to 2021. The increase in the number of SP-related publications followed a similar pattern as that observed in global scientific publications [30]. In addition, SP did not receive a lot of attention from scientists in the past. It is exciting that researchers in different countries have paid more and more attention to SP research, especially in the past 10 years (Figure 1a). This shows that there are many unsolved scientific problems in this field.

The average number of citations per publications has been increasing in the past 30 years (Figure 1b). One possible reason is that it takes longer time for new publications to reach the researchers, and older publications will gradually lose their novelty because people prefer to refer to recent publications. Another possible factor could be the correlation with the annual volume of journal publications [24]. The referencing behavior of scholars is inclined towards articles published in renowned journals that possess substantial impact. Hence, in order to enhance the visibility and recognition of their research, scholars focusing on SP should consider submitting their work to journals that boast high impact factors. We also found that most research fields were focused on plant science and agronomy (Figure 2). This may be due to the fact that SP is closely associated in the plant science and agronomy fields.

The seed is widely recognized as a crucial fundamental agricultural input for achieving enhanced crop yields. SP technology is a cost-effective and highly efficient method of seed pre-treatment that can enhance the quality of sowing materials [31]. The literature contains numerous reports on seed priming techniques aimed at enhancing germination uniformity, seedling emergence, stand establishment, crop growth, nodulation, and productivity in various crop species, such as maize [32,33], rice [34,35], wheat [36,37,38,39], alfalfa [40] and mung bean [41]. The priming procedures commonly employed in practice are typically proprietary and overseen by specialized agricultural seed company, e.g., EasyPrime, Emergis, and PROMOTOR^TM^ [6].

### 2.2. Related Journals

At the moment, there have been 983 journals that have published research on SP techniques, which indicates a significant distribution of SP publications across various journal scopes. As shown in Table 1, *Seed Science and Technology*, *Pakistan Journal of Botany*, *International Journal of Agriculture and Biology*, and *Horticulture* were the highly productive journals in the field of SP research. The aforementioned journals are specifically related to the field of plant science or agronomy and can be categorized as either professional or comprehensive in nature.

The top five journals with the highest number of citations in the field of SP studies were *Plant Physiology*, *Seed Science and Technology*, *Journal of experimental botany*, *Frontiers in Plant Science*, and *Plant and Soil* (Table 1). These analyses indicated that there is a positive correlation between the number of SP research papers published in journals and the corresponding citation count for SP studies. Furthermore, it is worth noting that journals with elevated impact factors frequently exhibit a greater number of citations for studies on SP, despite the limited amount of published research in this particular field.

### 2.3. Most Productive and Cited Authors

Through the analysis of the article’s author, we can utilize our advanced recognition capabilities to identify scholars who are actively engaged in research related to SP and have made a significant impact. Figure 3 displays the authors who have demonstrated exceptional productivity and those whose works have garnered significant citations, with a focus on the top 10 in each category. Out of the total 9441 authors, the ten most productive authors published 380 papers, accounting for 12.28% of the total publications. It is important to mention that we did not take into account the order of authorship in the calculation of publications and citations. Most of the authors began to pay attention to SP research in the last decade (Figure 4), which shows that SP research area has been attracting attention in the recent decade.

Figure 5 shows a network of productive authors. The author’s publication count is represented by the size of the circle, and the degree of collaboration can be inferred from the interconnection among writers. By analyzing the collaborative network among prolific authors, we can promptly identify the primary research teams in this specific field and which teams we mainly collaborate and learn from in this field. The collaboration between foreign scientists is relatively intensive. This may be one of the reasons why the number of citations of Chinese authors is not high, even though the number of published articles is large. Furthermore, the Mandarin-language publications by Chinese scholars has gone unnoticed.

### 2.4. Productive Organizations and Nations

The institution that is highlighted can serve as an indicator of the academic focus on SP research and aid in recognizing both the level of activity and influential institutions involved. During the course of this study, a total of 2843 global institutions actively participated in SP research. Notably, the top 10 distinguished institutions made a significant contribution with 1335 papers, representing over 43% of all publications examined. In the last three decades, the University of Agriculture Faisalabad has consistently held the top position in terms of publishing a significant (Table 2). Government College University and Islamic Azad University ranked second and third, respectively. It is worth mentioning that the majority of the leading productive institutions were situated in developing nations. The reason behind their focus on SP research was to improve seed vigor to increase crop productivity to achieve food security for large populations inhabiting these countries. Seed priming, which is known for its low-cost, has the potential to be efficiently utilized as a tactic by farmers with limited resources in order to enhance food security in extremely disadvantaged agricultural regions [20].

The country with the largest number of papers may reflect its emphasis on SP research in their national agriculture policy. The development of a comprehensive global heat map showcases the geographical distribution of research papers across different nations (Figure 6). Due to the comparatively lower level of technological development in the region, there is a relatively limited number of articles published, for example, in countries of Africa. The heat map enables the visualization of larger regions where the most productive or cited institutions and hotspots are located. Based on our research findings, SP-related studies have been conducted in a total of 89 countries worldwide (Figure 6). The number of papers contributed by corresponding authors from the top 10 countries is 2065 (66.72%), and the total number of citations from the top 10 countries with the highest citation counts is 48443. From 1991 to 2020, the top ten countries with the highest number of corresponding authored publications were India, USA, Pakistan, China, Iran, Brazil, UK, Turkey, Poland, and Italy. Nevertheless, the top 10 countries with the highest number of citations were USA, China, India, UK, France, Netherlands, Canada, and Australia (Figure 7). The level of collaboration among Pakistan, China, and Australia was quite high. However, it is necessary to enhance the level of international collaborations for these three countries in future research (Figure 8). This analysis allows us to identify variations in the quantity of SP publications and citations across various nations, and the need to promote cooperation between SP research and researchers from all over the world.

### 2.5. Temporal Evolution of Popular Keywords

#### 2.5.1. Keywords with the Highest Popularity

Keywords have the ability to showcase insights into current research directions and boundaries, along with highlighting the subjects that captivate researchers in this particular domain [30,42,43]. Out of a pool of 7071 keywords, the Wordcloud analysis revealed the top 50 frequently utilized keywords spanning over three decades (Figure 9). The keywords ‘seed priming’ or ‘priming’ were found to be the most commonly utilized, with a total frequency of 743. In addition to our research topic, the most studied topic word is “germination”. The whole intention of seed priming is to improve seed germination and uniform seedling establishment. Successful seed germination and uniform stand establishment is the premise of ensuring agricultural productivity. Meanwhile, the priming effect can be visually manifested through the germination characteristics, such as germination percentage, germination index, the mean period of ultimate germination, and synchronization index [44,45,46,47]. This may be the reason why germination is the most popular key word among the researchers considered in the study. The investigation of seed germination process has consistently remained a significant area within the field of seed science research, and improving the uniformity of seed germination is key to agricultural productivity.

The other important keywords that appeared in the publications were “Wheat” and “Salinity”(Figure 9). The analysis shows that 1212 research publications were on wheat seed priming. Thus, improving seed germination of wheat seeds through priming was a popular research area [37,38,39,48,49,50]. When it comes to the issue of salinity, it stands out as a primary abiotic stress that significantly impacts crop yield in arid and semi-arid areas. The germination of seeds can be negatively impacted by salt stress, leading to unfavorable physiological and biochemical alterations. It can affect seed germination and site through osmotic stress, the ion-specific effect, and oxidative stress. Salinity retards or prevents seed germination through various factors [51,52]. All kinds of techniques can improve the seedling emergence rate and stand establishment rate under salt conditions [53,54,55]. One of the most common methods is SP. The SP process involves prior exposure to non-living stressors, which enhances the seeds’ ability to withstand future exposure. The stress imprint induced by SP may contribute to the enhanced stress tolerance observed during post-priming germination, suggesting the presence of cross-tolerance mechanisms [14]. The key words of abiotic stress [5,13,18], salt stress [55], and drought stress [12,28,38,46] were also frequently mentioned. The seed triggers the metabolic process before germination to prepare the seed for radicle protrusion [21].

Currently, our findings indicate that the SP field primarily revolves around two main aspects, as depicted in Figure 10. The primary focus lies on investigating seed priming techniques and their impact on germination.

#### 2.5.2. Temporal Evolution of Keyword Frequencies

The research process has witnessed significant changes in the frequency of usage for all keywords in SP research over time. “Priming” has been a research hot spot in the past 30 years (Figure 11). From 1991 to 2000, the keywords classical agriculture research included “germination”, “seed priming”, and other crops, representing the primary areas of study within traditional agricultural practices (Figure 11). From 2001 to 2010, the research subjects remained consistent with the previous areas of focus. Following the year 2011, there was a notable upsurge in interest and emphasis on plant science and environmental science fields of study (Figure 11). The alterations in the primary research areas of SP over the last three decades were reflected through the modifications observed in the frequently utilized keywords. In the 1990s and the 2000s, researchers in the field of SP primarily focused on conducting extensive research in traditional agricultural studies, such as germination, seedling emergence, seed respiration, and storage after SP [56,57,58,59,60]. In the 2010s, their primary focus of research revolved around the fields of botanical science and environmental science. One possible reason could be that people have realized the necessity of gaining a deeper comprehension of the metabolic processes occurring during the priming treatment in order to utilize this technology more effectively. The terms ‘reactive oxygen species’ and ‘oxidative stress’ have become high-frequency words, which can reflect seed vitality, especially under stress conditions [61,62,63,64]. In the context of current global climate change, there is an urgent need to better understand the impact of climate change on SP, which is also an important topic [65]. In the past five years, new priming technologies have been under discussion, i.e., nanoparticles. Nanoparticles offer a promising strategy for regulating agricultural production systems due to their unique surface properties and small size [66,67].

## 3. Materials and Methods

### 3.1. Data Collection and Preparation

A bibliometric analysis requires a collection of large databases on SP research topics. We used the Web of Science (WOS) core collection in the WOS database (https://www.webofscience.com/WOS accessed on 14 November 2022) to establish a database on SP research. The utilization of WOS has gained significant recognition among researchers and has proven to be effective for conducting bibliometric analysis in recent research studies [29,30]. The SCI-E database is very comprehensive. It encompasses the globally impactful studies and incorporates easily accessible citation details, facilitating our ability to monitor the trajectory of SP research. Our research focuses on SP. In this section, the process of gathering and organizing data was primarily separated into two phases. The first stage was data retrieval. We searched the publications with the theme of “seed priming” from 1991 to 2021. We chose 1991 to 2021 as the research period, primarily due to the limited number of studies available on SP prior to 1991 in the database. We carried out bibliometric collection on 14 November 2022. We conducted an initial screening of the publications, focusing on 11 aspects, such as publication format, keywords, correspondence details, title, abstract content, citation count, publication year and date, volume and issue numbers, and digital object identifier. We implemented preprocessing techniques to identify instances of duplication and spelling errors. While the majority of bibliometric data is trustworthy, it is possible for the references provided to include various iterations of identical publications as well as variations in author names. The practice of authors using their surname and initials is prevalent, thereby presenting challenges in ascertaining their given names. Cited journals may also appear in different forms. This is an aspect that needs to be considered and checked in our analysis.

In our analysis, there were 3095 articles in total, mainly including proceeding papers, articles, comments and letters (Table 3). The second stage of data processing is downloading and converting data. For the subsequent bibliometric analysis, we transform the data using the ‘bibliometrix’ package in R software version 4.03.

### 3.2. Statistical Analysis

#### 3.2.1. General Statistical Analysis

This section primarily offers a comprehensive statistical analysis encompassing various aspects, including the quantification of articles published and referenced across different countries, journals, authors, and institutions.

#### 3.2.2. Most-Used Keyword Analysis

Frequently used keywords can well reflect research trends and frontiers [24]. Wordcloud has the ability to promptly display significant words to the reader sequence, enabling the assessment of their relative significance-associated with each word. We employed Wordcloud analysis to identify the top 50 frequently utilized keywords in SP research over the course of the study period. Each label signifies a specific expression, while varying font sizes and colors indicate distinct occurrences [24].

#### 3.2.3. Cooperation Networks Analysis

We analyzed the authors, keywords and countries cooperation networks. The frequency of the circle is indicated by its size. A larger circle corresponds to a higher frequency. The connection depicted by the line connecting two circular entities symbolizes the correlation existing between a pair of writers/terms/nations. The relationship between the two authors/keywords/countries becomes closer as the line thickness increases. Diverse color schemes are utilized to symbolize distinct clusters, suggesting a higher likelihood of these authors/keywords/countries being present in the same publication [24].

#### 3.2.4. Temporal Trend Analysis of Keywords

In order to gain a deeper understanding of how research topics have evolved over time, we conducted an analysis on the temporal trends. We categorized the publications in the 30 years into four distinct stages (1991–2000, 2001–2010, 2011–2015, and 2016–2021). By segmenting the span of 30 years into temporal intervals, the alluvial map can effectively depict the chronological progression of subjects within a particular domain of study [24]. All clusters consist of keywords, with each cluster comprising multiple keywords. These keywords represent multiple primary keywords within a cluster of related terms, and all clusters are arranged sequentially on the timeline. This timeline exclusively enlarges the keywords present in the cluster based on the specific year of their occurrence. Therefore, a timeline chart is generated.

In this study, R 4.0.3 software (utilizing the Bibliometrix package) and SigmaPlot 12.5 software (as shown in Figure 12) were employed for conducting bibliometric analysis, including statistical measurements and text analysis, such as annual publication and citation counts, and the identification of highly productive journals, authors, institutions, countries, and frequently used keywords.

## 4. Conclusions and Limitations

Based on bibliometric analysis, we comprehensively reviewed the papers published in the field of seed priming from 1991 to 2021. There has been a significant increase in SP research, along with a substantial rise in the annual average number for each paper. The most frequently used keywords were “priming” and “germination” in SP study. Over the past thirty years, *Seed Science and Technology*, *Pakistan Journal of Botany*, *International Journal of Agriculture and Biology*, *Horticulture* and *Seed science research* were the five most productive journals. However, *Plant Physiology*, *Seed Science and Technology*, *Journal of Experimental Botany*, *Frontiers in Plant Science,* and *Plant and Soil* ranked among the top five journals with the highest number of citations. The majority of the top 10 productive institutions were situated in developing nations, where they demonstrated exceptional research output with a significant number of publications and citations. These findings indicate that the significance of published papers lies primarily in their quality rather than the sheer quantity. We built the study trend of SP and the time evolution of popular keywords and revealed that agricultural science were the main topics in the 1990s and 2000s, and there was a trend toward plant science and environment science research in the 2010s. Our assessment of this paper was performed utilizing the SCI-E database, potentially leading to limited inclusion of the study as it excludes articles published in non-SCI journals and languages other than English. This should be enhanced in the future by incorporating additional databases. While the extent of search outcomes is restricted, our investigation offers an excellent initial reference for elucidating the temporal progression of SP. Our results show that SP research is limited in space, quantity and quality. Hence, it is imperative to enhance research advancements and foster collaboration, necessitating financial support and global coordination among scientists with diverse viewpoints and expertise.

## Figures and Tables

**Figure 1 plants-12-03483-f001:**
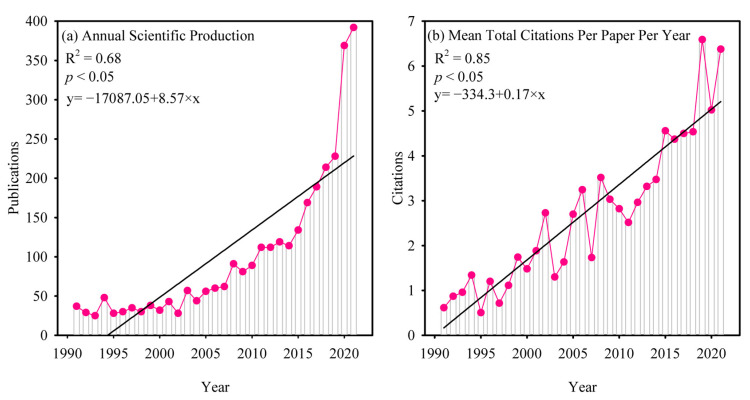
Relevant research on the study of seed priming (SP) has been conducted annually, spanning from 1991 to 2021, encompassing scientific publication output (**a**), in addition to the mean citation count obtained for each individual publication annually (**b**).

**Figure 2 plants-12-03483-f002:**
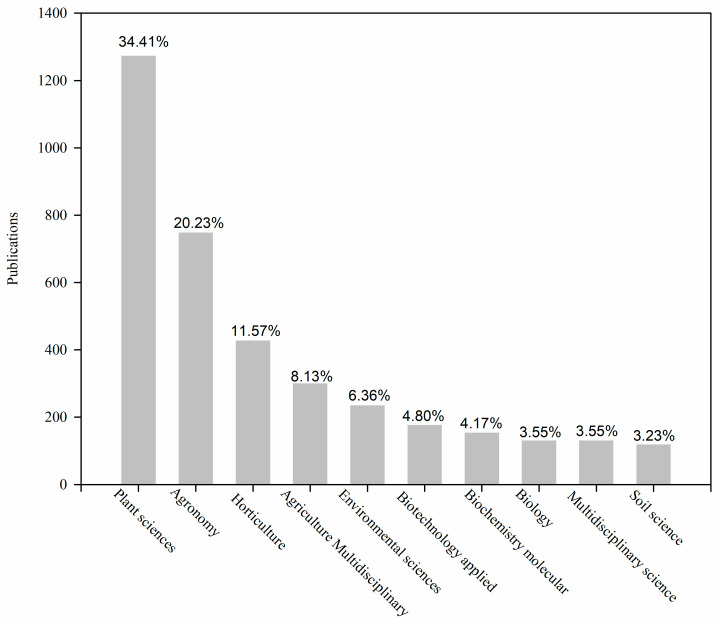
The total output of scientific publications of the most productive research areas on seed priming (SP) research from 1991 to 2021; the research areas are classed by the Web of Science. The numbers on each bar chart represent the percentage of publications in the research field.

**Figure 3 plants-12-03483-f003:**
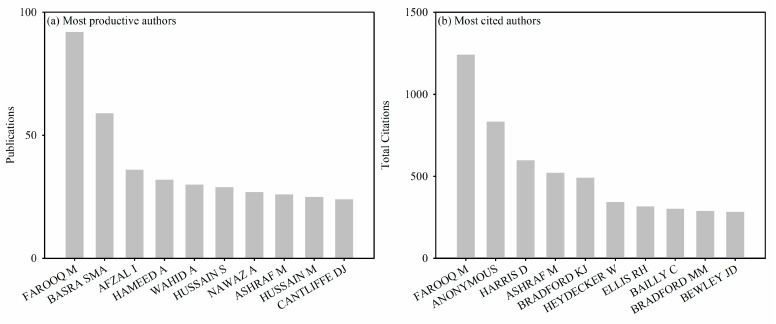
(**a**) Authors with the highest level of productivity and (**b**) authors with the highest number of citations among the top ten.

**Figure 4 plants-12-03483-f004:**
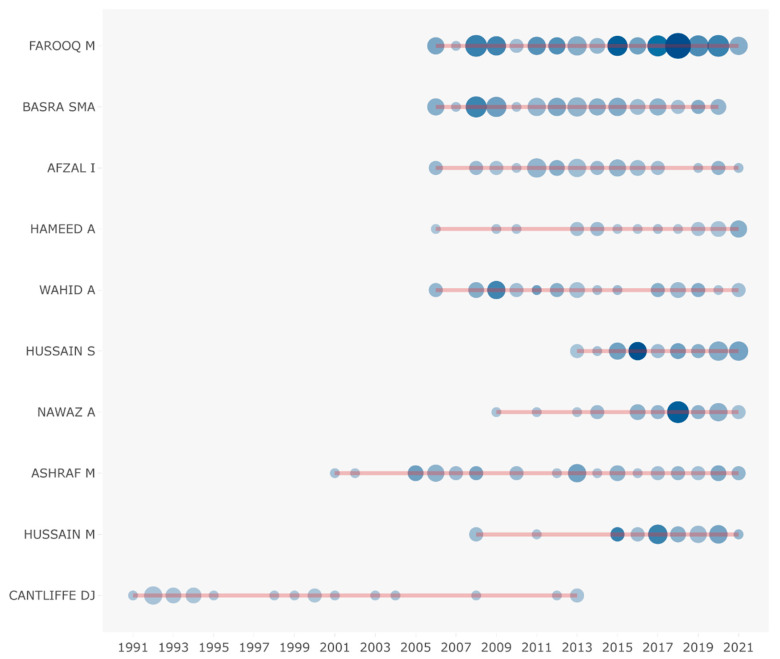
Evolution of the number of articles of the top ten most productive authors from 1991 to 2020.

**Figure 5 plants-12-03483-f005:**
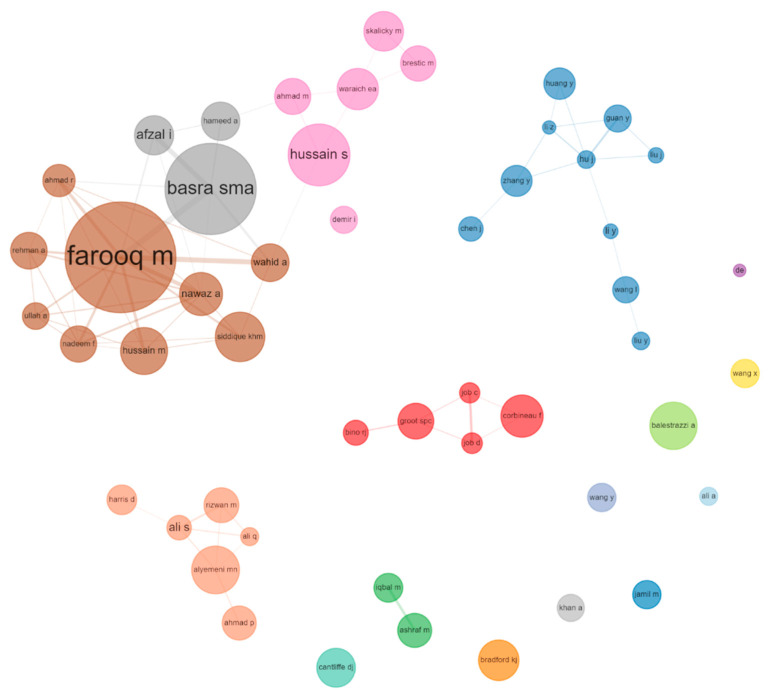
Network depicting the co-occurrence of the 50 most prominent authors. The frequency of the author is indicated by the size of the circle, with larger circles indicating higher frequencies. The connections depicted by the lines connecting two circles represent the associations between two authors, with the thickness of the line indicating the proximity of their relationship. Different clusters are represented by various colors, suggesting a higher occurrence of these authors in shared publications.

**Figure 6 plants-12-03483-f006:**
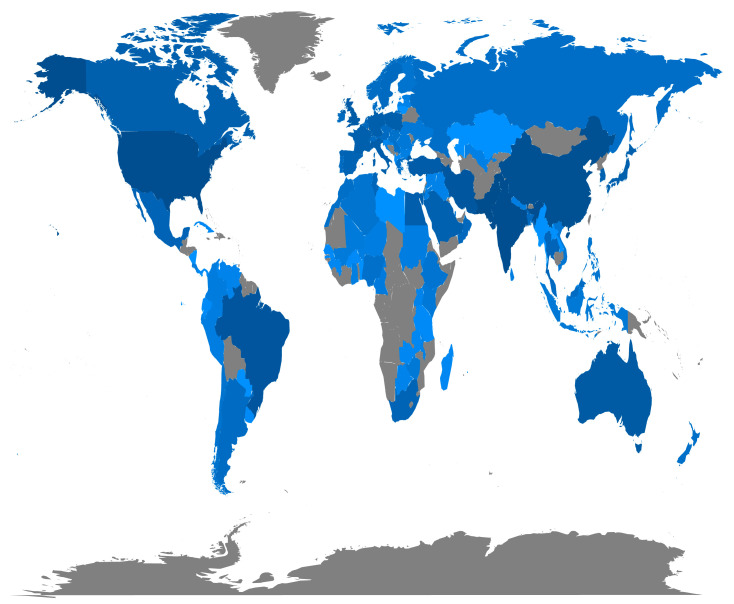
Scientific output of nations. The level of color saturation corresponds to the quantity of research papers on seed priming (SP) investigations.

**Figure 7 plants-12-03483-f007:**
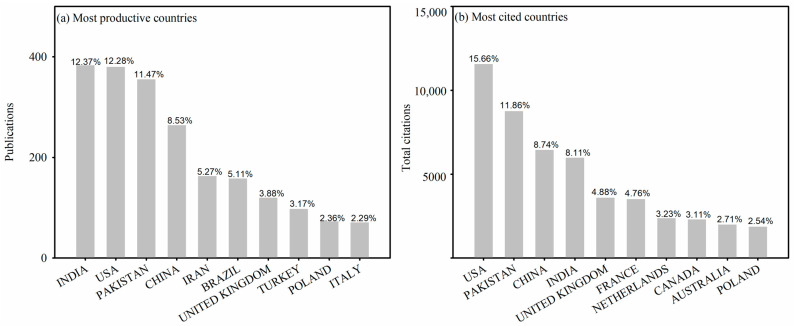
(**a**) Top ten most productive countries and (**b**) most citied countries. The number on each bar chart represent the percentage of publications or citations in the research field.

**Figure 8 plants-12-03483-f008:**
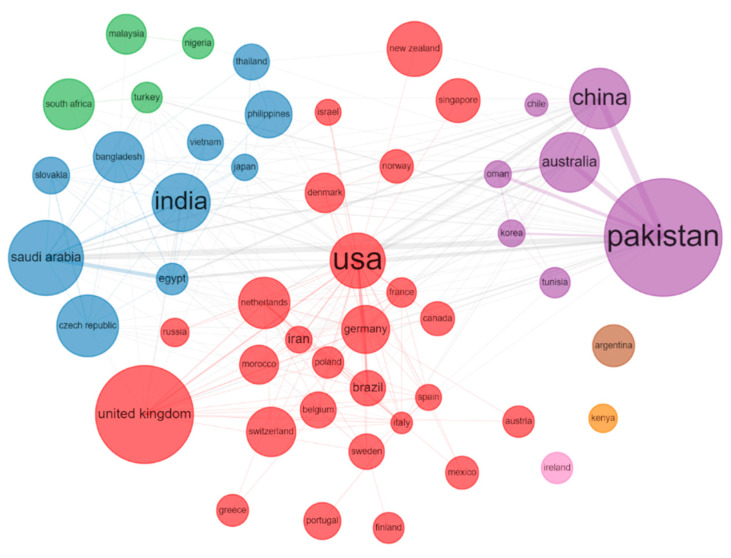
Network depicting the co-occurrence among the leading 50 nations. The frequency of the country is indicated by the magnitude of the circle, and the frequency increases as the size of the circle grows. The connections depicted by the intersecting lines of two circles represent the interrelations between a pair of nations, and the relationship between two countries becomes closer as the thickness of the line increases. Different clusters are represented by various colors, suggesting that these countries tend to appear together more frequently in publications.

**Figure 9 plants-12-03483-f009:**
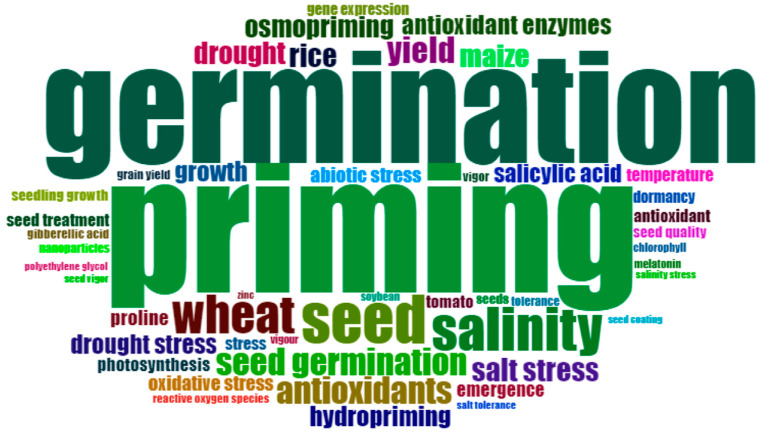
The Wordcloud presents the frequently employed keywords within the top 50. The size of the word represent the frequency of the word.

**Figure 10 plants-12-03483-f010:**
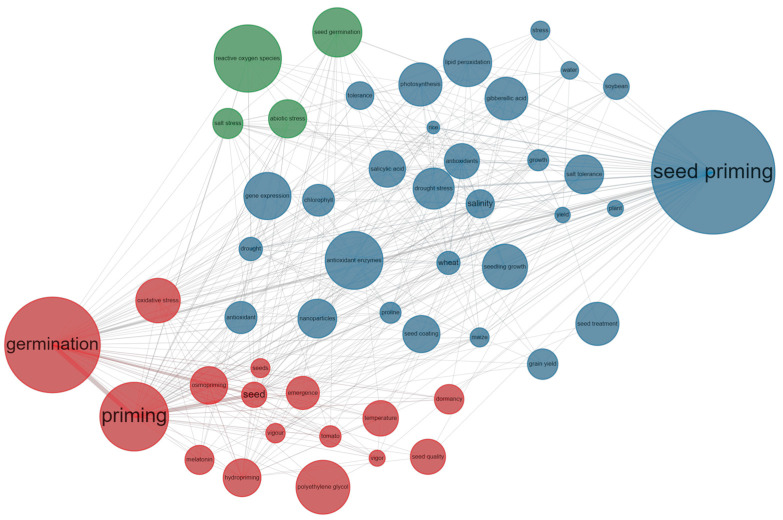
Network depicting the co-occurrence of the frequently used 50 keywords. The frequency of the keywords is indicated by the size of the circle, with larger circles indicating higher frequencies. The connections between two circles represent the associations between two keywords; the thickness of the line signifies the proximity of the relationship between them. Distinct clusters are denoted by varying colors, signifying a higher frequency of occurrence for these keywords within the same publication.

**Figure 11 plants-12-03483-f011:**
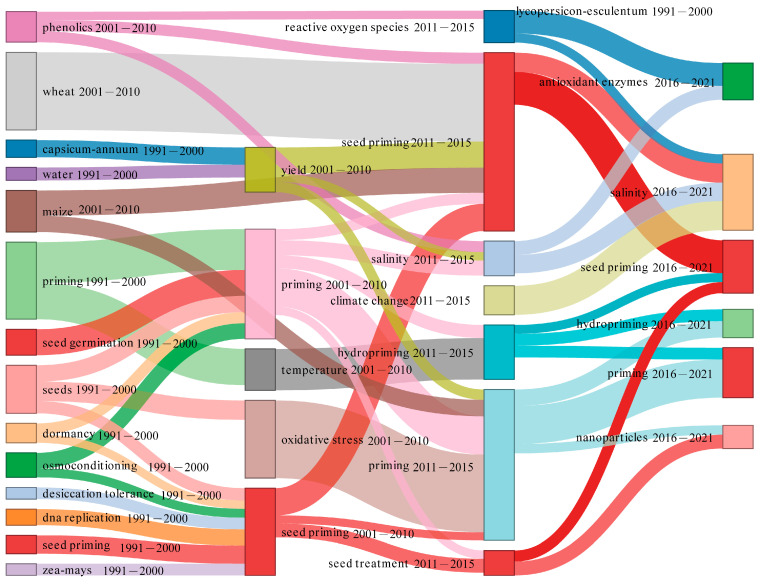
The temporal evolution of frequently employed terminology in the field of seed priming investigation. Every individual node symbolizes a frequently used keyword, with the dimensions of each node being directly proportional to its rate. The connection between each node illustrates the chronological progression of the keyword. These lines depict the correlation between keyword transmission and inheritability. Different colors represent different key words.

**Figure 12 plants-12-03483-f012:**
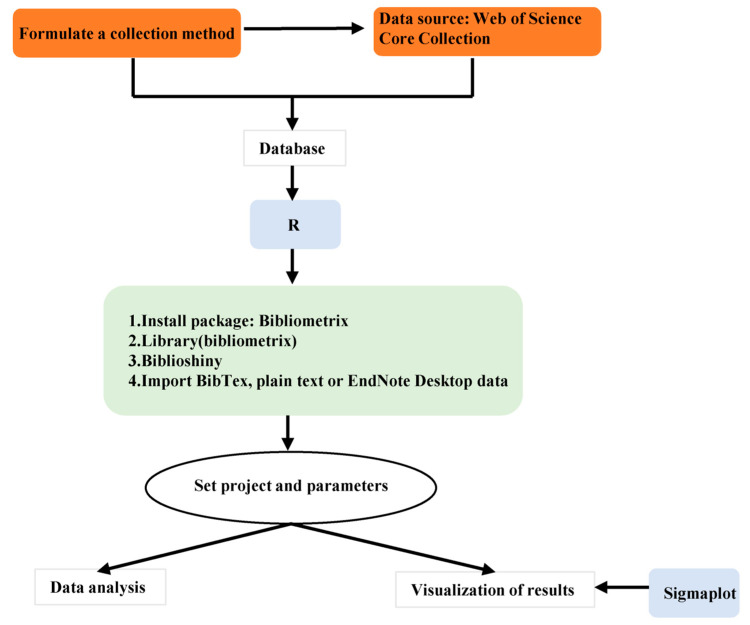
R bibliometric analysis flowchart.

**Table 1 plants-12-03483-t001:** List of the 20 most prominent and frequently referenced journals that have published studies on seed priming (SP) from 1991 to 2021.

Journal	Publications	H- Index	Journal	Total Citations
*Seed Science And Technology*	145	24	*Plant Physiology*	4115
*Pakistan Journal of Botany*	66	14	*Seed Science and Technology*	2888
*International Journal of Agriculture and Biology*	51	14	*Journal of experimental botany*	2764
*Horticulture*	49	16	*Frontiers in Plant Science*	1818
*Seed science research*	48	27	*Plant and Soil*	1481
*Journal of Plant Nutrition*	45	12	*Physiologia plantarum*	1465
*Plant Physiology and Biochemistry*	43	23	*Annals of Botany*	1437
*Frontiers in Plant Science*	42	24	*Plant physiology and biotechnology*	1404
*Acta Physiologiae Plantarum*	39	22	*Seed science research*	1384
*Legume Research*	36	5	*PNAS*	1380
*Agronomy-Basel*	35	12	*Journal of Plant Physiology*	1347
*Scientia Horticulturae*	35	16	*Plant science*	1269
*Scientific Reports*	34	21	*Crop science*	1252
*PLOS One*	32	16	*Environmental and Experimental Botany*	1210
*Environmental and Experimental Botany*	30	16	*Plant growth regulation*	1208
*Plants-Basel*	30	13	*Horticulture*	1180
*Indian Journal of Agricultural Sciences*	29	4	*Nature*	1147
*Journal of Agronomy and Crop Science*	28	19	*Scintia Horticulturae*	1056
*Journal of Plant Growth Regulation*	28	14	*Acta Physiologiae Plantarum*	1015
*Plant growth regulation*	24	16	*PLOS One*	1011

**Table 2 plants-12-03483-t002:** Top ten most marked institutions.

Institution	Publications
University of Agriculture Faisalabad	532
Government College University	135
Islamic Azad University	107
Bahauddin Zakariya University	103
University of Western Australia	102
King Saud University	86
Zhejiang University	75
Indian Agriculture Research Institute	66
Huazhong agriculture university	65

**Table 3 plants-12-03483-t003:** Main information regarding the collection.

Descriptions	Results
Timespan	1991–2021
Number of sources (journals, books, etc.)	983
Number of documents	3095
Total number of keywords	7071
Total number of authors	9441
Annual years from publication	9.13
Average citations per documents	23.86
Number of author appearances	13560
Number of single authored documents	119
Number of multi authored documents	9322
Number of documents per author	0.328
Number of authors per document	3.05
Number of co-authors per documents	4.38
Collaboration Index	3.15

Note: Collaboration index = publications co-authored by multiple individuals.

## Data Availability

All relevant data for this study are reported in this article.

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
