# Peer review of "Trends in Seed Priming Research in the Past 30 Years Based on Bibliometric Analysis"

_plants, 2023, doi:10.3390/plants12193483_

Round 1

Reviewer 1 Report

In the present manuscript, a bibliometric analysis based on the Science Citation Index-Expanded (SCI-E) database regarding the use of seed priming i agriculture was conducted.

The text should be restructed and does not have to follow the structure of a research article. Therefore, Results and Discussion are not relevant to a review manuscript.

Figure 1: The horizontal axis in 1a needs more values instead of only 200 and 400. Add some more values in between (e.g. 25, 50, 75 etc.). The same applies for 1b.

Figure 2. It is not clear which are the years where this figure refers to.

Figure 7: correct the caption (citied to cited).

In Table 1, it would be interesting to add one column for the range that each hjournal is active and one more column for the evolution of its impact factor. This could justify the low quality of publications, mentioned in Line 359. Similarly, the h-index and the citations they received for papers related to SP could also show the quality of published papers.

Reviewer 2 Report

This document is a review of seed-priming publications in the last three decades. The manuscript is well-written, with minor points to fix. I added some comments that may be helpful for the authors to increase the discussion and may get more visibility:

**Title: OK

**Abstract: OK

**Keywords: The keywords are good, but I recommend adding “International collaboration”.

1. Introduction

-Paragraph 2 (lines 44-55): I recommend adding more information about the seed priming (SP) process. The authors commented that there are many different types of SP. If readers get more information about the diversity of priming types, the document may be more visible and increase the odds of citation.

2. Results and Discussion

2.1. Temporal Trends of Publications and Citations

-Figure 1b looks like a linear trend, but Figure 1a is more like a power trend. Please consider reanalyzing the number of publications per year as a power relation.

-Please add the equation generated by the data analyses in Figure 1.

-Figure 2: The bar chart looks like a PCA scree plot, and I suggest adding the percentage of the publications on or in the bar for each research area. Furthermore, please add the y-axis: “Publications.”

-It is essential to discuss the most common trends, but I would like to ask the authors to discuss the less common trends. For example, Biochemistry and Biotechnology are the 5th and 6th most common research areas. If SP improves germination by changing the transcriptomics and metabolomics, why do these areas lack more research? Please hypothesize on this big gap between “Plant Sciences” and other research areas, commenting on how to boost science in other areas.

-It may be too late to ask this, but if the authors can, please indicate a number (for example, five or six) of the most commonly primed cultures. The authors comment that maize, rice, wheat (monocotyledon crops), alfalfa, and mung bean (from the Fabaceae family) are common primed crops. Why are they primed? Is it possible to increase the number of publications on SP for other crops?

2.2. Related Journals

-Please, if possible, write in the section the number of articles published as open access and subscription access per time stage (1991 – 2000, 2001 – 2010, 2011 – 2015, and 2016 – 2021) to detect if there is a trend of replacing subscription access by open access. This information may be helpful to justify increments in the budget for article publication by policymakers.

2.3. Most productive and cited authors

-Figure 3: If the articles agree, I would like to suggest formatting the bar chart as Figure 2 (no outline, blue filling) to keep visual consistency. Furthermore, please add the y-axis for Figure 3b: “Total citations.”

2.4. Productive Institutions and Countries

-Line 175: Please delete one of the double periods.

-Figure 7: If the articles agree, I would like to suggest formatting the bar chart as figure 2 (no outline, blue filling) to keep visual consistency. Furthermore, I suggest adding the percentage of the publications/total citations on or in the bar for each research area.

-Please comment on the low number of publishing countries in Africa.

2.5. Temporal Evolution of Popular Keywords 

2.5.1. Most Popular Keywords

-Line 217: It is unclear what the “it” means. Please replace “it” with the appropriate noun. Would it be “germination,” please?

2.5.2. Temporal Evolution of Keyword Frequencies: OK

3. Materials and Methods:

-If possible, I STRONGLY advise the authors to present the “Materials and Methods” section before the “Results and Discussion” section. There is an explanation about the graphs that must appear before the graphs. 

3.1 Data Collection and Preparation: OK

3.2 Statistical Analysis 

3.2.1 General statistical analysis:

-Please briefly describe the type of general statistical analysis. Did the authors use means, medians, quantiles, or other general statistical analyses?

3.2.2 Most used keyword analysis: OK

3.2.3 Cooperation networks analysis: OK

3.2.4 Temporal trend analysis of keywords: OK

4. Conclusions and Limitations: OK

**Author Contributions: OK

**Funding: OK

**Data Availability Statement: The authors generated data for this review. They created the BibTex, plain text or EndNote Desktop data, and the scripts for R. Therefore, there must be a Data Availability Statement in the document. Please check https://www.mdpi.com/journal/plants/instructions#suppmaterials and select the most suitable statement.

**Conflicts of Interest: OK

**References: 

-Reference 8: Please check if the page number is missing (page 37?).

-Reference 10: Please check if the volume and pages are missing (Pagano, A., Macovei, A. & Balestrazzi, A. Molecular dynamics of seed priming at the crossroads between basic and applied research. Plant Cell Rep 42, 657–688 (2023). https://doi.org/10.1007/s00299-023-02988-w).

-Reference 15: Please check if the volume and pages are missing (Tian Y, Guan B. Responses of Seed Germination, Seedling Growth, and Seed Yield Traits to Seed Pretreatment in Maize (Zea mays L.). Scientific World Journal 2014, 2014: 834630. DOI: 10.1155/2014/834630?).

-Reference 18: Please check if the page number is missing (page 8101?).

-Reference 36: Please check if the page number is missing (page 107404?).

-Reference 37: Please check if the page number is missing (page 93?).

-Reference 38: Please check if the page number is missing (page 5544?).

-Reference 39: Please check if the page number is missing (page 1052660?).

-Reference 41: Please check if the page number is missing (page 2006?).

-Reference 42: Please check if the page number is missing (page 104798?).

Reviewer 3 Report

The manuscript is well presented and quite interesting. However, there was no attempt to access the quality of the paper published, The fact that certain country or author has the most publications does not indicate its real value in contribution to the scientific community. Some minor revisions are needed. On Line 50 delete "since times"; Line 117 delete "and et al" and insert "and" before "mungbean"; Line 239 delete "and et al" and insert "and" before "synchronization"

The English is acceptable. Minor revisions have been mentioned in previous section.
